# Study on the Control of Dichloroacetonitrile Generation by Two-Point Influent Activated Carbon-Quartz Sand Biofilter

**DOI:** 10.3390/membranes12020137

**Published:** 2022-01-24

**Authors:** Xinrui Gui, Huining Zhang, Bixiao Ji, Jianqing Ma, Meijuan Xu, Yan Li, Ming Yan

**Affiliations:** 1Polytechnic Institute, Zhejiang University, Hangzhou 310000, China; 21960557@zju.edu.cn; 2School of Civil Engineering & Architecture, NingboTech University, Ningbo 315100, China; JiBixiao@nit.zju.edu.cn (B.J.); majq@nbt.edu.cn (J.M.); xmj80@126.com (M.X.); liyanliyan@nbt.edu.cn (Y.L.); 3Ningbo Research Institute, Zhejiang University, Ningbo 315100, China; 4Ninghai Branch of Ningbo Ecological Environment Bureau, Ningbo 315600, China; yanming7727@126.com

**Keywords:** activated carbon-quartz sand biofilter, nitrogen-containing disinfection by-products, Dichloroacetonitrile, Tyrosine, drinking water

## Abstract

Aiming at the problem of highly toxic Nitrogenous disinfection by-products (N-DBPs) produced by disinfection in the process of drinking water, two-point influent activated carbon-quartz sand biofilter, activated carbon-quartz sand biofilter, and quartz sand biofilter are selected. This study takes typical N-DBPs Dichloroacetonitrile (DCAN) as the research object and aromatic amino acid Tyrosine (Tyr), an important precursor of DCAN, as the model precursor. By measuring the changes of conventional pollutants in different biofilters, and the changes of Tyr, the output DCAN formation potential of the biofilters, this article investigates the control of DCAN generation of the two-point influent activated carbon-quartz sand biofilter. The results show that the average Tyr removal rate of the three biofilters during steady operation is 73%, 50%, and 20%, respectively, while the average effluent DCAN generation potential removal rate is 78%, 52%, and 23%, respectively. The two-point influent activated carbon-sand biofilter features the highest removal rate. The two-point water intake improves the hypoxia problem of the lower filter material of the activated carbon-quartz sand biofilter, and at the same time, the soluble microbial products produced by microbial metabolism can be reduced by an appropriate carbon sand ratio, which is better than traditional quartz sand filters and activated carbon-quartz sand biofilters in the performance of controlling the precursors of N-DBPs.

## 1. Introduction

The disinfectant used in the process of drinking water disinfection will react with most natural organic matter as well as producing harmful Disinfection by-products (DBPs) [1]. DCAN, as a typical N-DBPs, has higher cytotoxicity and genetic toxicity compared with other common DBPs. It has been detected many times in the disinfection process and has attracted attention [2,3]; as a result, DCAN was selected as the research object in the experiment. The control of N-DBPs can be divided into source control, process control, and end control, in which source control is an effective control way [4]. Dissolved organic nitrogen (DON) is the main precursor of N-DBPs [5], and the source control of N-DBPs is mainly to remove DON. Compared with natural organics, DON usually has the characteristics of low molecular weight and low electrostatic charge. The removal effect of conventional treatment processes (coagulation, flocculation, sedimentation, and sand filtration) is limited. Srithep et al. [6] evaluated the performance of conventional treatment processes in removing Haloacetonitriles (HANs) precursors and found that they can remove about 28% of DON, but cannot effectively remove HANs precursors. Cuthbertson et al. [7] found that if biofiltration was used before disinfection, N-DBPs precursors could be effectively removed. The water quality analysis results from Xiang et al. [8] showed that the biofilter process has a good effect on the removal of pollutants, especially organic micro pollutants. Therefore, the biofilter process is selected as the focus on source control research.

In recent years, the application of biological filtration in the advanced treatment of drinking water has gradually risen. The effectiveness of biologically active filtration for denitrification and the removal of DBPs precursors has been confirmed in most studies [9], but the current research on the removal of N-DBPs precursors is still in its infancy. The traditional quartz sand filter is difficult to meet the drinking water treatment standards under the condition of the micro pollution of the water source. Moreover, the biological activated carbon process has the problem of microbial leakage; domestic and foreign studies show that the activated carbon-quartz sand biofilter derived from the two can remove organic matter well. In view of this, the activated carbon-quartz sand biofilter is selected to control the generation of DCAN. Zhang et al. [10] showed that the activated carbon-quartz sand biofilter had the best control effect on DON when the activated carbon-quartz sand ratio was 2:8. At the same time, it was found that two-point influent can control the DON concentration more effectively than single-point influent, which is why the two-point influent activated carbon-quartz sand biofilter with activated carbon-quartz sand ratio of 2:8 was selected in the experiment. Amino acid (AA) is an important component of DON in water, accounting for 35% of DON [11]. Some components of DON, such as aromatic AA, are easy to generate N-DBPs and difficult to remove. Therefore, Tyr, an important precursor of DCAN, is used as a simulated precursor to analyze its changes in the biofilter and explore the control of the biofilter on the formation of DCAN. Due to the high toxicity and commonness of DCAN, the widespread existence of Tyr as an important precursor and some physical characteristics, it is difficult to effectively remove it in the conventional process. Therefore, if the Tyr DCAN precursor can be effectively removed before disinfection, so as to reduce the formation of DCAN, it has a very far-reaching practical significance. At the same time, it also lays a theoretical foundation for the study of the biofilter control of N-DBPs.

## 2. Materials and Methods

### 2.1. Test Device

The test device is shown in Figure 1. The main structure is made of plexiglass, with an inner diameter of 70 mm and a height of 2000 mm. The biofilter includes a water inlet pipe, a water outlet pipe, a backwash water pipe, a supporting layer, and a filter media layer. The 200-mm-high supporting layer is made of gravel. With particle sizes of 0.5~1 mm, and 1~2 mm, respectively, the filler of the filter layer is quartz sand and coconut shell activated carbon. The packing layer of filter A is 1000 mm quartz sand. The packing layers of filter B and C are both the upper layer of 200 mm coconut shell activated carbon and the lower layer of 800 mm quartz sand. According to the design specification of the water plant, the filtration rate of the filter is 8 m/h, and the influent flow rate is 30 L/h. Filter A and B are single-point influent, and filter C is two-point influent, and the two influent points are located at the top and 20 cm depth from the filter, respectively; the first influent point flow rate is 20 L/h, and the second influent point flow rate is 10 L/h. Filter A and B are the blank control of filter C. The biofilter is backwashed every 24 h.

### 2.2. Inoculated Sludge and Influent Water Quality

As the biofilter adopts the method of inoculation and film hanging, the inoculated sludge is taken from the secondary sedimentation tank of the municipal sewage plant. Each filter column is inoculated with 1 L sludge, and the sludge concentration during inoculation is 2 g MLVSS/L. The experiment makes use of artificial water distribution, and the water quality of the experiment refers to the effluent from the sedimentation tank of The Ningbo Water Plant. Gu et al. [12] found that the concentration of DON in the source water of Zhejiang, China was 0.9~1.8 mg/L. Therefore, the dosage of Tyr was selected as 10 mg/L so that the DON content in the mixed water was 0.61~1.03 mg/L. The test water quality is shown in Table 1.

### 2.3. Analysis Methods

Collect influent and effluent samples of three biofilters every day. The water sample is filtered with 0.45 μm fiber membrane before analysis, and each water sample is measured 3 times. NH_4_^+^-N is measured by Nessler’s reagent photometric method; NO_2_^−^-N is measured by N-(1-naphthyl)-ethylenediamine photometric method; NO_3_^−^-N is measured by UV spectrophotometry; Total nitrogen (TN) is measured by alkaline potassium persulfate digestion UV spectrophotometry; Chemical oxygen demand (COD) is measured by standard potassium dichromate method; Dissolved oxygen (DO) and pH value are measured by portable analyzer; turbidity is measured by turbidity meter. The biomass is determined by the phospholipid method [13] and the calculation formula of DON is as shown in Equation (1).
DON = TN-NH_4_^+^-N-NO_2_^−^-N-NO_3_^−^-N(1)

### 2.4. Determination of Tyrosine Content

Tyr analysis is measured by Hitachi L8900 automatic amino acid analyzer. Put 50 mL of the water sample into a centrifuge tube, pretreat the water sample with a rotary evaporator, concentrate it for 10 times, inject the mixed amino acid standard working solution and sample determination solution of the same volume into the amino acid automatic analyzer, and calculate the Tyr concentration in the sample determination solution by the external standard method through the peak area.

### 2.5. Determination of Formation Potential of DCAN

The DCAN formation potential is measured by a fully chlorinated method [14], Shimadzu GC-2030 gas chromatograph is used, sodium hypochlorite solution is acted as a disinfectant, and the effective chlorine dosage is as shown in Equation (2).
*C* = 3*C*_1_ + 7.6*C*_2_ + 10(2)

Among them: *C*—Cl_2_ dosage, mg/L;

*C*_1_—DOC, mgC/L;

*C*_2_—NH_3_ concentration, mgN/L.

Fully mix after chlorination, adjust the solution to pH = 7 with NaHCO_3_ buffer, store it at 25 °C away from light and at a constant temperature for 24 h, and then add ascorbic acid to remove the residual chlorine in the water sample. Then take 50 mL of water sample with a 50 mL centrifuge tube, add 5 g of anhydrous sodium sulfate, put it on the vibrator for full shaking for 1 min, add 5 mL of extractant methyl tert butyl ether for liquid-liquid extraction and enrich DCAN in the water sample. The chromatographic column was used for DCAN detection. The formation potential of DCAN was measured by the modified EPA551.1 method. The sample inlet is 200 °C, the detector is 290 °C, the temperature rise procedure is 35 °C, hold for 10 min, then raise the temperature to 145 °C at 10 °C/min for 2 min, and then raise the temperature to 260 °C at 20 °C/min for 5 min.

### 2.6. Microbial Community Analysis

Microorganisms were analyzed by high-throughput sequencing [15]. On the 148th day of the experiment, samples were taken from the sampling ports at different filtration depths of three biofilters.

## 3. Results and Discussion

### 3.1. Analysis of the Removal Effect of Biofilter Pollutants

#### 3.1.1. Removal Effect of Conventional Pollutants

During the test period, Figure 2a shows the change of NH_4_^+^-N concentration on the inlet and outlet water of the biofilter. The effluent NH_4_^+^-N concentration of the biofilter gradually stabilizes in about 30 days, and the effluent NH_4_^+^-N concentration is between 0.14 and 0.47 mg/L, while the NH_4_^+^-N removal rate is above 60%. The average removal rate of TN in the biofilter is shown in Figure 2b. The average removal rate of TN in the three filters: Filter C (28%) > Filter B (21%) > Filter A (12%) (*p* < 0.05), and filter C possesses the best nitrogen removal effect. Denitrification produces N_2_, organic nitrogen or inorganic nitrogen, which is used and transformed by microorganisms into N element in microbial cells for microbial growth and reproduction, which are all possible ways to remove TN.

After filtration, the turbidity of the effluent from the three filters was significantly reduced (Figure 2c), and the removal effects of filters B and C were better than those of filter A, which is consistent with the previous research conclusions [16,17]. The turbidity of filter C is the lowest, which can meet 80%, indicating that the two-point influent method not only shows no impact on the removal of turbidity but also can further enhance the turbidity removal effect.

#### 3.1.2. Removal Effect of Tyrosine and COD

The average removal rates of Tyr and COD_Mn_ in the effluent of the biofilter are shown in Figure 3. During the experiment, the concentration of Tyr in the effluent of the three filters decreased as follows: Filter C (73%) > Filter B (50%) > Filter A (20%) (*p* < 0.05). The Tyr concentration of biofilter effluent is lower than that of influent, which is inconsistent with the conclusion made by Liu et al. [18], who recorded the increase in DON concentration in the effluent after biofiltration in the drinking water treatment plant, indicating that the change of DON overall effluent concentration does not represent the change of individuals, and the change law of different individuals in the filter tank is also different. The change trend of average effluent COD_Mn_ removal rate is consistent with that of average effluent Tyr removal rate. The COD_Mn_ removal rate can reflect the removal of organic matter in the biofilter. Tyr is an organic matter, and its molecular structure contains reducing phenolic hydroxyl and amino groups, which can be oxidized by potassium permanganate. Therefore, the COD_Mn_ removal rate can also be applied as the basis of Tyr removal. Meanwhile, except for the influent mode, other conditions of filter B and C are the same, but the organic matter removal effect of filter C is significantly better than that of filter B, indicating that the two-point influent improves the organic matter removal rate of biofilter.

### 3.2. The Formation Potential and Influencing Factors of DCAN in the Effluent of the Biofilter

The formation potential for DCAN in the influent and effluent of the biofilter is shown in Figure 4a. During the experiment, the formation potential of effluent DCAN was relatively stable, and the formation potential of effluent DCAN decreased to varying degrees after the three filter tanks. The formation potential of DCAN in the effluent of filter C is the lowest, indicating that it has the best effect on removing DCAN precursors. The generation potential for DCAN in the effluent of the three filters lessens from the Tyr concentration. Tyr is a crucial precursor of DCAN [19], and the removal of Tyr can cut down the generation of DCAN in the subsequent disinfection process. The formation potential for DCAN is related to many factors, such as backwashing and operating conditions of the filter [20,21]. Liu et al. [22] found that Soluble microbial products (SMPs) released by bacterial metabolism are considered to be the primary source of DON in drinking water biofilters. The research of Zhou et al. [23] confirmed that NH_4_^+^-N, NO_3_^−^-N and NO_2_^−^-N in water are also important sources of nitrogen in N-DBPs. Therefore, the SMP_S_ produced by the internal microbial action of the biofilter and other organic matter in the effluent and even NO_3_^−^-N and NO_2_^−^-N may be the precursors of DCAN, affecting the effluent DCAN formation potential.

Backwashing will also impact the formation potential for DCAN in the effluent of biofilter. Figure 4b shows the changes in the effluent DCAN formation potential with backwashing time. During continuous operation within one backwash cycle (24 h), the effluent DCAN formation potential is lower than that of the influent water. When the continuous operation time exceeds 24 h, the formation potential for DCAN in the effluent of the three filters increased significantly and was higher than that of the influent. Among them, the growth of filters B and C was larger, and the formation potential for DCAN in the effluent exceeded that of filter A. The reason for this phenomenon is that under long-term continuous operation, the adsorption reaches saturation and cannot play a role, the adsorbed pollutants are gradually released, and the SMP_S_ produced by microorganisms will accumulate and release. The adsorption performance of activated carbon is stronger than that of quartz sand, and the number and activity of microorganisms in the activated carbon-quartz sand biofilter are also stronger than that of quartz sand biofilter. Therefore, the impact of continuous operation on activated carbon-quartz sand biofilter is higher than that of quartz sand biofilter.

### 3.3. Analysis of Changes in Biofilter along the Way

The changing rules of Tyr concentration along the biofilter are shown in Figure 5a. The Tyr concentration of filters A and B first decreased rapidly and then increased slowly, and this is consistent with the conclusion that the concentration of DON in the biofilter first plunges and then rises slowly. Many studies have shown that the DON concentration first decreases and then increases in the biofilter. For example, Liu et al. [18] found that the DON concentration decreases at 0~10 cm and increases at 10~200 cm. Zhang et al. [24] divided the DON concentration change into 0~20 cm decrease stage and 20~100 cm increase stage. Liew et al. [25] believe that this is because with the increase in biofilter depth, the biomass and microbial activity of filter materials at different depths first increase and then decrease. Among them, the concentration of Tyr in filter C increased briefly because of the inflow of water at 20 cm, and then dropped to 50 cm and crept up. The decrease rate of Tyr concentration on the activated carbon layer of filters B and C is much greater than that of filter A; it may be that the activated carbon has a large specific surface area, strong absorption capacity, and more attached microorganisms [26], while the smooth surface of quartz sand is not conducive for microorganisms to attach. The Tyr concentration of filter A is basically unchanged between 30~100 cm, while that of filter B rises significantly after 50 cm, and the increase in filter C in the second half of the filter is smaller than that of filter B. The three filters consume a lot of oxygen in the upper half of the filter material (as is shown in Figure 5b), so the amount of oxygen distributed in the lower layer is reduced, which reduces the number of microorganisms in the lower layer, and the microbial decomposition of Tyr in the lower layer is limited. Previous studies have shown that Tyr is a typical representative of SMP_S_ released by bacteria [22], so the accumulation of SMP_S_ is also the reason for the increase in Tyr concentration in the lower layer of the filter. The influent of filter C at the depth of 20 cm brings in DO, which increases the DO at 20 cm depth (as is shown in Figure 5b), supplements the DO of the lower filter material, improves the microbial environment to a certain extent, and enables microorganisms to play a better role. The rate of decline of filter C in the 20~50 cm stage is greater than that of filter B, while the rise in the 50~100 cm is slower than that of filter B. The biomass of the biofilter changes along the filter material layer depth as shown in Figure 5(c), and the biomass decreases together with the increase in the filter layer depth. The biomass of filter A decreased rapidly, which may be because the surface of the quartz sand is not conducive to the growth of microorganisms. Except for the depth of 0~10 cm, the biomass of filter B and C at the depth of other filter materials is greater than that of filter A. The possible reason is that although the activated carbon has a huge specific surface area, it is mostly concentrated in micropores and is not fully utilized by aerobic microorganisms. Therefore, the biomass of activated carbon layer of filter B and C at 0~10 cm is smaller than that of the quartz sand layer of filter A. In the second half of the filter, the microorganisms in filter B and C that are not intercepted by activated carbon are intercepted by quartz sand, so the biomass decreases slowly. The biomass of filter C increased temporarily due to the inflow at the depth of 20 cm.

### 3.4. Conversion of Nitrogen in the Biofilter

Nitrogen in drinking raw water mainly exists in the form of NH_4_^+^-N, NO_2_^−^-N, NO_3_^−^-N, and organic nitrogen [27]. Most of the organic nitrogen in this experiment is Tyr. In the biofilter, dissolved inorganic nitrogen (DIN) and DON are mutually converted, so it is necessary to discuss the conversion to nitrogen in the biofilter. The changes of nitrogen in the water in and out of the biofilter are shown in Figure 6. The DIN loss is Filter C > Filter B > Filter A (*p* < 0.05), which is consistent with the removal rate of COD_Mn_. The utilization of organic matter by the biofilter increases with the increase of DIN loss, indicating that the DIN loss phenomenon in the reaction is related to filter denitrification [28].

The reactions related to nitrogen transformation in the biofilter mainly include nitrification, denitrification, ammoniation, and so on. The oxygen in the upper part of the filter is sufficient, and the nitrification reaction is mainly carried out in the upper part of the filter. The oxygen in the lower part of the filter is insufficient, and the denitrification reaction is mainly carried out in the lower part of the filter. The ammoniation of Tyr degradation can occur in all stages along the biofilter, and the reaction generates ammonia nitrogen to realize the transformation from organic nitrogen to inorganic nitrogen. Denitrification produces N_2_, and microbial growth in the filter consumes some N elements, which will cause DIN loss. At the same time, organic nitrogen conversion will make up for DIN loss to a certain extent. There is a certain linear correlation between the variation of Tyr in the effluent of the three biofilters and the nitrogen loss in the filter, and the correlation coefficients of the biofilters are: Filter A (0.44), Filter B (0.63) and Filter C (0.73). The correlation coefficients of biofilters are between 0.3 and 0.8, indicating that there is a certain correlation between the loss of DIN and the change of Tyr in the three biofilters. Some scholars have confirmed that the amount of DIN loss is positively correlated with NH_4_^+^-N concentration, and increases with the increase of NH_4_^+^-N concentration [29]. From the correlation coefficient, it can be known that the loss of DIN in this experiment is positively correlated with the change of Tyr, so there is a certain relationship between the change of Tyr and the concentration of NH_4_^+^-N. it is speculated that Tyr will be degraded by ammoniation in the biofilter to produce NH_4_^+^-N. It is confirmed from the side that part of Tyr is removed by ammoniation. In the meantime, it is only correlated but not strongly correlated, indicating that Tyr can also be removed by other means. Among the three types of filters, filter C has the strongest correlation, and filter A has the weakest correlation.

### 3.5. Analysis of the Microbial Community in the Biofilter

#### 3.5.1. Species Diversity Analysis

It’s shown in Table 2 for the species diversity index of samples from different biofilters and different filter layer heights. The larger the Shannon index, the higher the community diversity. The larger the Chao index, the higher the species richness. At different filter depths, the Shannon index and Chao index of filter C are higher than those of other filters, indicating that its species diversity and richness are the highest. Previous studies have shown that higher microbial diversity and richness are conducive to microbial degradation [30,31], which also explains why the degradation rate of organic matter in filter C and the removal rate of Tyr are the highest.

#### 3.5.2. Analysis of Microbial Community Structure at the Phylum Level

The microbial community structure of the sample of the phylum level is shown in Figure 7. A total of seven main phyla were detected in samples of different biofilters, namely *Proteobacteria*, *Planctomycetes*, *Bacteroidetes*, *Firmicutes*, *Chloroflexi*, *Actinobacteria*, and *Chlorobi*, among which, *Proteobacteria* can use organic matter as nutrition [32], participating in the degradation of a series of multifunctional aromatic protein substances [33]. *Bacteroidetes* is also considered to be very important to the degradation process of organic matter [34]. Studies have shown that *Actinobacteria* can also remove nitrogen-containing organic matter, so it is speculated that they are related to the degradation and removal of Tyr. The bacterial phylum involved in the removal of Tyr in the sample is discussed and analyzed separately, as shown in Figure 8. The changes along the way in abundance of bacteria in the Tyr-related phylum of the three filters are consistent with the changes in the Tyr along the way in the figure, indicating that the removal of Tyr is closely related to the microorganisms in the biofilter. By comparing the abundance changes along the way between filter B and filter C, the two-point influent has improved the phenomenon that the bacterial abundance related to the removal of Tyr in the activated carbon-quartz sand biofilter decreases significantly in the lower half of the filter, which is also the reason why the Tyr concentration of filter B increases significantly in the lower half. At the same time, the abundance of bacteria related to Tyr removal is evenly distributed in the filter, which ensures the stability and high efficiency of Tyr removal.

## 4. Conclusions

The two-point influent activated carbon-quartz sand biofilter is superior to other filters in removing conventional pollutants (NH_4_^+^-N and turbidity) and organic pollutants (COD_Mn_ and Tyr); the average NH_4_^+^-N removal rate is 80%, turbidity is 80%, COD_Mn_ is 49%, and Tyr is 73%. The two-point influent activated carbon-quartz sand biofilter has the best control effect on the formation of DCAN. The removal of Tyr in the biofilter is concentrated on 0~30 cm in the upper part of the filter. DO decreases along the upper part of the filter, and then stabilizes, and the total biomass of the two-point influent charcoal-sand biofilter is the largest. The correlation coefficient between DIN loss and Tyr change in the two-point influent activated carbon-quartz sand biofilter is 0.73, which possesses the highest correlation, and its biochemical property is the highest, which is beneficial to the removal of precursors. The total abundance of bacteria related to Tyr removal in two-point influent activated carbon-quartz sand biofilter is the highest among the three filters.

## Figures and Tables

**Figure 1 membranes-12-00137-f001:**
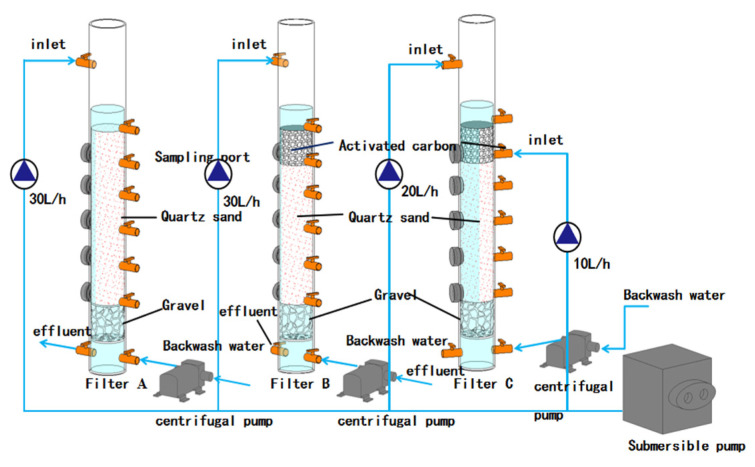
Operating device.

**Figure 2 membranes-12-00137-f002:**
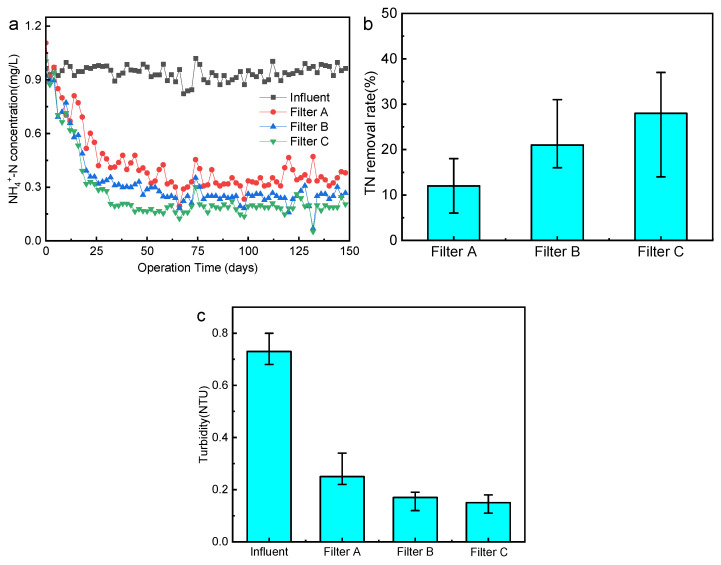
Biofilter (**a**) NH_4_^+^-N concentration in influent and effluent (*n* = 3); (**b**) the average removal rate of TN (*n* = 3); (**c**) the turbidity of influent and effluent (*n* = 3).

**Figure 3 membranes-12-00137-f003:**
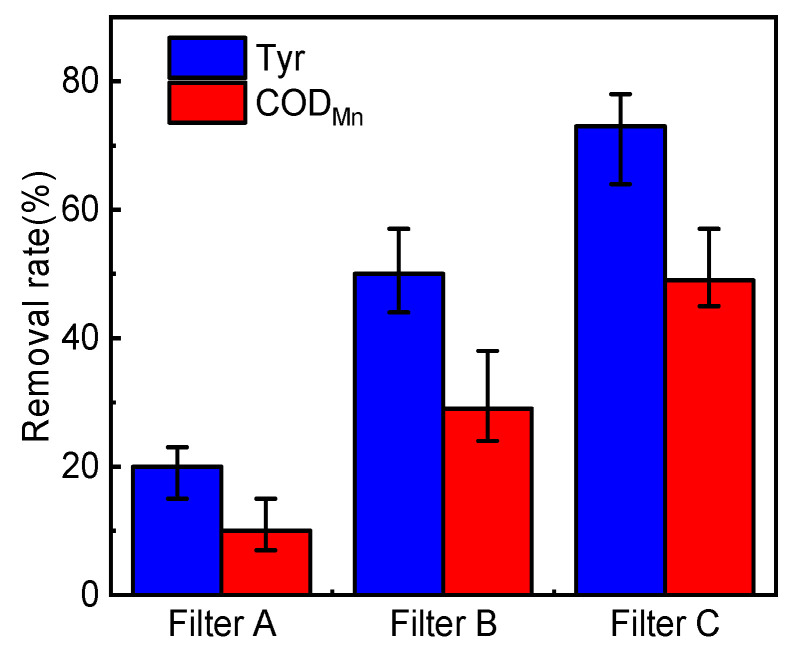
Average removal rates of Tyr and COD_Mn_ in the biofilter effluent (*n* = 3).

**Figure 4 membranes-12-00137-f004:**
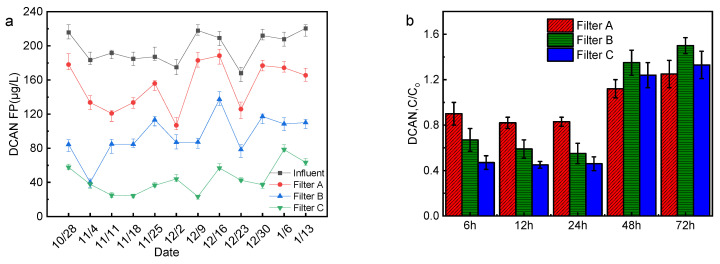
DCAN formation potential of biofilter (**a**) Influent and effluent (*n* = 3); (**b**) Changes with backwash time (*n* = 3).

**Figure 5 membranes-12-00137-f005:**
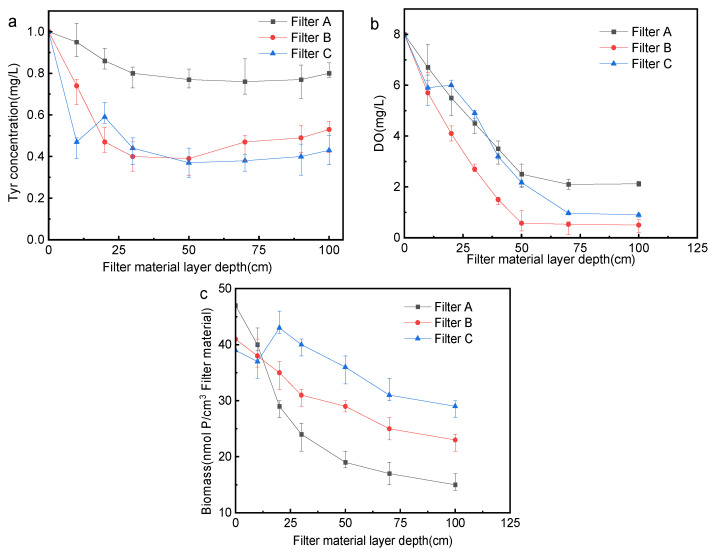
Changes of biofilter along filter material layer depth (**a**) Tyr (*n* = 3); (**b**) DO (*n* = 3); (**c**) Biomass (*n* = 3).

**Figure 6 membranes-12-00137-f006:**
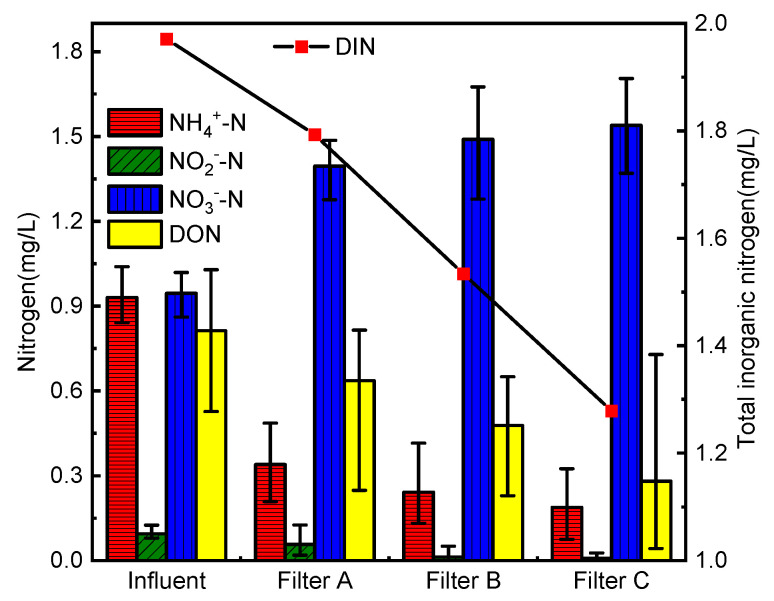
Changes of nitrogen in influent and effluent of biofilter (*n* = 3).

**Figure 7 membranes-12-00137-f007:**
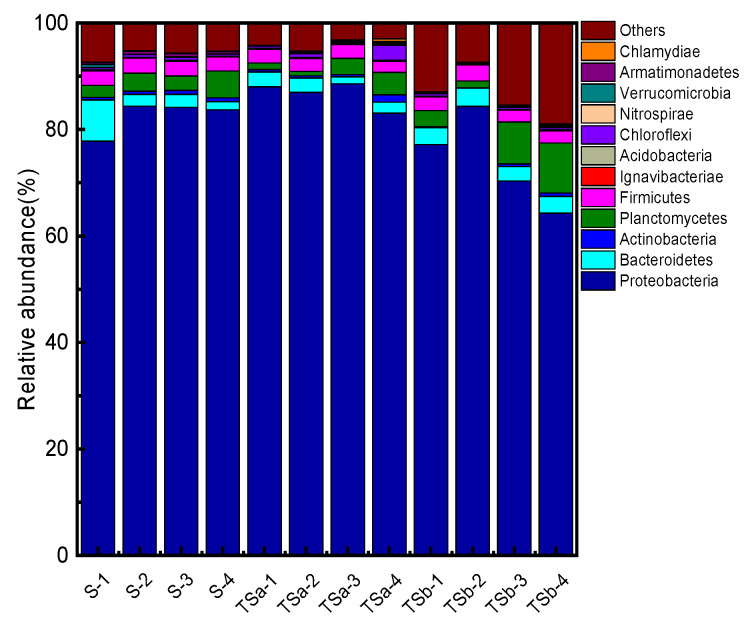
Horizontal community structure distribution of bacterial phyla in biofilter samples.

**Figure 8 membranes-12-00137-f008:**
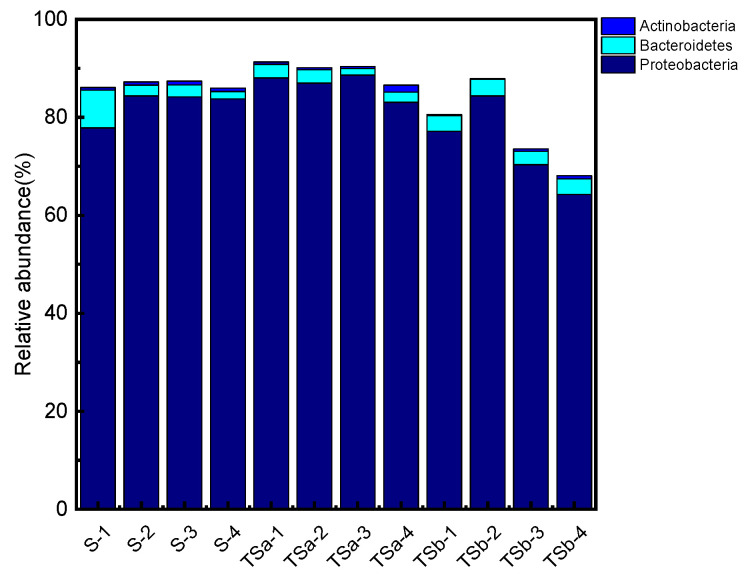
Removal of Tyr-related microbiota.

**Table 1 membranes-12-00137-t001:** Influent characteristics during experiment.

Index	NH_4_^+^-N/(mg·L^−1^)	NO_2_^−^-N/(mg·L^−1^)	NO_3_^−^-N/(mg·L^−1^)	pH	DO/(mg·L^−1^)	DON/(mg·L^−1^)
Content	0.82~1.02	0.083~0.125	0.87~1.08	7.01~8.94	6.67~9.53	0.61~1.03

**Table 2 membranes-12-00137-t002:** Alpha diversity index of biofilter samples.

Sample	Shannon Index	Chao1 Index	Good’s Coverage
S-1	2.64	675.56	1
S-2	2.39	616.13	1
S-3	2.49	716.13	1
S-4	2.28	606.36	1
TSa-1	2.85	611.37	1
TSa-2	3.15	788.4	1
TSa-3	2.91	697.33	1
TSa-4	3.77	774.23	1
TSb-1	2.82	584.03	1
TSb-2	2.3	522.67	1
TSb-3	2.74	649.96	1
TSb-4	3.14	680.51	1

Note: S-1, S-2, S-3, and S-4 indicate the filter material of the quartz sand biofilters at a distance of 10, 20, 50, and 100 cm from the surface; TSa-1, TSa-2, TSa-3, and TSa-4 represent the filter layer material at a distance of 10, 20, 50, and 100 cm from the surface of the two-point influent activated carbon-quartz sand biofilter; TSb-1, TSb-2, TSb-3, and TSb-4 represent the distance of the activated carbon-quartz sand biofilter. The filter material of the filter layer at the surface of 10, 20, 50, and 100 cm, and the rest of the table is the same as the figure.

## Data Availability

The data presented in this study are available on request from the corresponding author.

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
