# Peer review of "Study on the Control of Dichloroacetonitrile Generation by Two-Point Influent Activated Carbon-Quartz Sand Biofilter"

_membranes, 2022, doi:10.3390/membranes12020137_

Round 1
Reviewer 1 Report
The study shown in the paper 'Research on Controlling DCAN Generation in Dual-point Water Inlet Activated carbon-quartz sand Biofilter ' is interesting. The research presented in the paper can be used by other scientists and water treatment system operators . The chapter 'Introduction' requires some rewrite. There is no reference to other filtration systems. In the discussion chapter, there is no comparison of own results with the results of other researchers. I found the paper to be well organized and written.
Author Response
Thank you very much for your comments and suggestions, we have made changes one by one according to your suggestions. Please see the attachment.

Reviewer 2 Report
I read the paper and I suggest major revisions before a possible publication. My comments are the following:
- Please, carefully read the text and amend the grammar errors. Please, avoid too long sentences (e.g., lines 40-43). Please, rephrase them.
- In the introduction, more detailed about the existent literature on this topic and current challenges in this sector should be added.
- Moreover, the final part of the Introduction what are the main aspects studied in this work should be briefly reported, highlighting the novelty points with respect to existent literature. Why your study is important in this sector? Try, to answer these aspects.
- Tables (e.g. table 1) should be placed after the reference in the text.
- Section 2.3. The references of the methods should be provided in references list.
- Please avoid “formula” and use “equation”.
- Section 2.3-2.4-2.5-2.6 should be merged.
- Section 2.6. Please, provide essential info of the methodical procedure used.
- When a confidence interval is provided (e.g., in figures), please, provide also the number of samples used to determine the interval (e.g., in the caption).
- Section 3.1., please, results should be only reported but also discussed and compared with previous literature results
- Figures 4a and 5. Are confidence intervals available? If yes, please, insert them.
- Figure 7 is very difficult to read. Please, try to insert 7a and 7b as two separate figures.
- In conclusion, practical implications of your findings should be added.
- I suggest to insert a nomenclature. It will be helpful due to the high number of abbreviations.
Author Response

(The authors gave the same response as above.)

Reviewer 3 Report
Please see my comments in the attached file

Author Response

(The authors gave the same response as above.)

Round 2
Reviewer 2 Report
In my opinion, the work can now be published.
Reviewer 3 Report
Recommending to accept in present form